# Targeting CD24 in Cancer Immunotherapy

**DOI:** 10.3390/biomedicines11123159

**Published:** 2023-11-27

**Authors:** Wenwen Chen, Zhigang Hu, Zhigang Guo

**Affiliations:** Jiangsu Key Laboratory for Molecular and Medical Biotechnology, College of Life Sciences, Nanjing Normal University, 1 Wenyuan Road, Nanjing 210023, China; 221202068@njnu.edu.cn

**Keywords:** CD24, siglec-10, immunotherapy

## Abstract

Immunotherapy is a hot area in cancer treatment, and one of the keys to this therapy is the identification of the right tumour-associated or tumour-specific antigen. Cluster of differentiation 24 (CD24) is an emerging tumour-associated antigen that is commonly and highly expressed in various tumours. In addition, CD24 is associated with several cancer-related signalling pathways and closely interacts with other molecules and immune cells to influence tumour progression. Monoclonal antibodies, antibody–drug conjugates (ADCs), chimeric antigen receptor (CAR) T-cell therapy, and CAR-NK cell therapy are currently available for the treatment of CD24. In this review, we summarise the existing therapeutic approaches and possible future directions targeting CD24.

## 1. Introduction

CD24 (cluster of differentiation 24), also referred to as nectadrin, is a glycoprotein anchored by glycosylphosphatidylinositol (GPI). Although mature CD24 has a short protein core containing only 31 to 34 amino acids, it has an extensive N-linked and O-linked glycosylated structure [1]. In 1978, mouse heat-stable antigen (HSA), a homologue of human CD24, was discovered and named after its heat resistance [2].

Over time, it has become evident that CD24 contributes to numerous inflammatory, immune, and neurological diseases. Recently, a study [3] pinpointed CD24 as an anti-phagocytic cell surface protein functioning as a “don’t eat me” signal. Consequently, cancer cells exhibiting CD24 surface expression are capable of evading immune cell surveillance and restrict the immune response via this mechanism. CD24 has been identified as a promising target for cancer immunotherapy, and targeting it with antibody drugs could offer effective treatment options.

This review builds on existing research on CD24, exploring its structure and role in cancer, with a focus on current studies of antibody drugs targeting the protein and potential treatments. The findings could pave the way for innovative approaches to cancer treatment in the future.

## 2. Structure

The *Cd24* gene in mice resides on chromosome 10 [4], while the *Cd24* gene in humans is situated on chromosome 6q21 [5]. The length of the human *CD24* gene is 80 amino acids. Like the human *CD24*, the mouse *Cd24* mRNA encodes a small peptide that may contain only 30 amino acids upon removal of the signal sequence and replacement of the C-terminal region by the glycosyl-phosphatidylinositol group [6]. Most of the weight of human CD24 (HSA) is due to the considerable N- and O-linked glycosylation at various locations on the short peptide [6]. Figure 1 illustrates the anticipated fundamental structure of CD24. Earlier studies have indicated that the glycosylation of HSA and CD24 is exceedingly varied and depends on the cell type [7]. The glycoform of CD24 differs in expression in diverse cells or tissues, which may impact the multiple functions of CD24 [8]. Furthermore, it is noteworthy that the CD24 structure in cancer cells differs from that of normal cells, and CD24 in cancer cells may be highly glycosylated [9]. C Ohl conducted a comparative analysis on N-glycosylation of CD24/HSA from diverse sources, and highlighted similarities and differences between them [8]. Unfortunately, there is currently no research that explicitly displays the precise structure of CD24-related glycans. Hence, it is worth considering whether designing CD24 antibody drugs from the viewpoint of CD24 glycans is a feasible approach.

## 3. Expression Pattern of CD24

### 3.1. CD24 Is Widely Expressed in Cancer

CD24 is highly expressed in various tissues under physiological conditions, specifically the oesophagus, pancreas, and thyroid. Additionally, it is widely expressed in haematopoietic cells, with a high expression observed in B cells, as well as non-haematopoietic cells and various cancer cell types, such as ovarian cancer, breast cancer, colorectal cancer, lung cancer, and other haematological and solid tumours (Figure 2). In comparison with other innate immune checkpoints, CD24 expression is extensively upregulated in diverse cancer cells [3]. Therefore, targeting CD24 is meaningful in treating various tumours with high CD24 expression and poor prognosis. However, it is crucial to consider ways to prevent harm to normal tissues during treatment. It is worth noting that CD24 is not expressed in human erythrocytes, so targeted CD24 therapy does not result in anaemia as a side effect [10].

CD24 is generally considered to be present on the plasma membrane of cells, and its localisation is associated with lipid rafts. Nevertheless, various accounts indicate that CD24 may also be found intracellularly, presumably located in vesicles (see https://www.proteinatlas.org/ENSG00000272398-CD24/subcellular#human accessed on 10 October 2033). Intracellular CD24 is usually linked to cancer and has a distinctive function [11]. CD24 immunostaining of colon tissue revealed that the tumour epithelium’s staining pattern comprised the plasma membrane, cytoplasm and nucleus. Normal epithelial cells and non-epithelial cells showed no nuclear staining [12]. Similar findings were confirmed in hepatocellular carcinoma, with CD24 protein demonstrating cytoplasmic immunostaining in HCC tumour cells and negative expression in neighbouring non-tumorigenic liver tissue [13]. Keisuke Taniuchi discovered that intracellular CD24 can interact with phosphorylation-dependent endonuclease G3BP to inhibit the invasiveness and metastasis of cancer cells. This is achieved by altering G3BP RNase activity, which influences the post-transcriptional regulation of BART mRNA levels [14]. CD24 is crucial in the advancement of bladder cancer metastasis, and the implementation of CD24 short hairpin RNA in bladder cancer cells leads to an almost complete termination of the tumour cells’ metastatic ability. In contrast, surface CD24-negative (surCD24^−^) cells only exhibit a 50% decrease in metastatic ability [15]. Jason E Duex and colleagues have demonstrated that surCD24^−^ cells in bladder cancer contain CD24 residues in the nucleoplasm, and that it is the nucleoplasmic CD24 (nucCD24) that is responsible for driving surCD24^−^ growth and metastasis [16]. The entry of nucCD24 into the nucleus leads to binding with nuclear chromatin, although the significance of this binding remains unclear. In Lizhong Wang’s study, it was suggested that there may be a certain relationship between nucCD24 and the binding of nucleolar protein NPM [17]. Although the function of nucCD24 remains unknown, it is a crucial component that requires attention.

### 3.2. CD24 Is Closely Related to Cancer Stem Cells

A small number of cells in various types of cancer display features similar to those of stem cells, and these cells are commonly referred to as cancer stem cells (CSCs) [18]. Tumours are hierarchical tissues composed of heterogeneous cell populations driven by cancer stem cells (CSCs) [19]. CSCs are multipotent and self-regenerating [18]. There is indication that these exceptionally tumorigenic cells may possess the ability to metastasise [20]. Additionally, CSCs have been observed to demonstrate resistance to chemotherapy and radiotherapy [18]. Thus, cancer stem cells (CSCs) are a subgroup of cancer cells accountable for the initial tumour formation, its maintenance, and recurrence. CSC targeting holds promise as a potential therapy for cancer. CD24 has been identified as a significant marker for tumour diagnosis and prognosis [21]. In pancreatic cancer, CSCs have been identified with the phenotype CD24^+^ [18]. However, in breast cancer, the CD44^+^/CD24^−^ cell population, also known as “breast CSCs,” exhibits considerably greater tumorigenic capability than the CD44^+^/CD24^+^ cell population [22]. Given the close correlation between CSCs and CD24, CD24 emerges as a promising target for the treatment of cancer initiation and recurrence [21]. Nevertheless, this research section also demonstrates noteworthy differences in CD24 expression among various cancer cell lines. The complexity of CSC markers renders the identification of CSCs across different cancer types impossible with a single marker. Thus, further research and evidence are required to comprehend this topic fully [23].

## 4. The Function of CD24 in Cancer Progression

### 4.1. Signalling Pathways Associated with CD24 and Cancer

#### 4.1.1. Src-Mediated Signalling Pathway

Protein tyrosine kinase Src, particularly c-Src, is often overexpressed in a range of cancers and can be a significant factor in tumour progression [24,25]. The connection between CD24 and Src in relation to their location and function has been gradually recognised [26,27] (Figure 3). The methods by which CD24 supports tumour progression via Src are explained as follows.

Firstly, CD24 impacts the motion of cancerous cells in two distinct manners: firstly, through an integrin-dependent mechanism that encourages adhesion to substrates, and, secondly, through an integrin-independent mechanism that enhances cell motility [28]. The movement and infiltration of malignant tumours typically necessitate cell adhesion to the extracellular matrix via β1 heterodimeric integrin receptors [29]. Within the undamaged lipid rafts of breast cancer cells, CD24 interacts with c-Src, enhancing its activity as opposed to modifying the expression level of c-Src. Boosted Src activity accelerates the phosphorylation of its substrates FAK and paxillin, consequently increasing integrin-mediated adhesion. This indirectly promotes cell adhesion to fibronectin, collagen I and IV, and laminin, through the activation of α3β1 and α4β1 integrin, which in turn fosters cancer cell metastasis and invasion [30]. However, not all β1 integrins co-localise with CD24, and cancer cells do not solely enhance their invasive capacity through the integrin-dependent pathway described above. P-Selectin is present on the surface of activated platelets and endothelial cells, and binds to ligands on myeloid and lymphocyte subpopulations [31]. CD24 was discovered to be a ligand for P-selectin in mouse bone marrow and endothelial cells in 1994 [27]. In human cancer cells, the interaction between CD24 and P-selectin facilitates the rolling of specific cancer cells on endothelial cells. This also leads to their adhesion to platelets and mesothelial cells, while promoting migration and spread of cancer cells, including breast cancer [32] and ovarian cancers [33].

Secondly, CD24 may enhance the proliferation and survival of cancer cells through the Src/STAT3 pathway. As mentioned above, CD24 interacts directly with Src in lipid rafts to activate it. Niko P. Bretz scrutinised CD24 expression in A549 lung cancer, SKOV3ip ovarian cancer, and HS683 and SNB19 glioblastoma cells, and discovered that Src, activated by CD24 tyrosine, phosphorylates STAT3 signalling molecules, which causes the activation of STAT3. Upon activation, STAT3 forms dimers and translocates towards the nucleus, where it specifically binds to DNA, promoting the expression of target genes. CD24 enhances the upregulation of STAT3-dependent genes, such as cyclin D1, survival protein, and MCL-1, thus promoting survival and proliferation of cancer cells [26]. Furthermore, it has been observed that there is a slight downregulation of STAT3 expression [26]. However, reports indicate that CD24 silencing does not downregulate Src in certain cancer cell lines, like SKBR-3, and actually upregulates Src in others, like MDA-MB-468. This could be due to lower levels of integrin β1 and CD44 [34].

#### 4.1.2. Epidermal-Growth-Factor-Receptor-Mediated Signalling Pathway

The study identified a significant increase in expression levels of epidermal growth factor receptor (EGFR) in CD24^+^ colorectal cancer (CRC) cells [35]. The correlation was further confirmed via clinical data of gastric cancer [36] (Figure 3). The subsequent investigation verified CD24’s inhibition of EGFR internalisation and degradation via RhoA in SGC-7901 and BGC-823 gastric cancer cell lines [37]. EGFR upregulation is a crucial process in gastric cancer cell metastasis [37]. The PI3K/Akt and ERK pathways are downstream pathways of EGFR that are vital to cancer cell growth and migration [38]. Akt and/or ERK activation can downregulate E-calmodulin and promote cancer cell migration [39,40].

#### 4.1.3. Ras/Ral-GTPases-Mediated Signalling Pathways

There is a mounting body of evidence indicating that heightened Ras activity and/or expression is linked to an enhanced likelihood of developing cancer [41]. CD24 is similarly expressed across a range of cancers, with an emerging correlation with Ras. In cases of non-small cell lung cancer (NSCLC), CD24 was identified as a downstream protein of Ras, and active Ras was found to inhibit CD24 expression [42] (Figure 3). Ras-like GTPases (Ral GTPases) are another class of small G proteins that are associated with the regulation of cancer cell morphology and motility [43]. In bladder cancer cells, Ral GTPases, like Ras, influence the regulation of CD24 [44].

#### 4.1.4. Mitogen-Activated-Protein-Kinase-Mediated Signalling Pathway

Mitogen-activated protein kinases (MAPKs) are intracellular serine/threonine protein kinases. The MAPK cascade reaction has an impact on cell growth, differentiation, and apoptosis. Mammalian MAPKs are classified into three categories: ERK, JUK, and p38MAPK [45]. CD24 overexpression induced SW480 cells’ proliferation. Weifei Wang also observed MAPK pathway activation, in which Weifei Wang also detected activation of the MAPK pathway. For the first time, it has been discovered that CD24-triggered growth relies on ERK1/2 and p38 MAPK activation in CRC. Its effect has no bearing on JUK-induced proliferation (Figure 3).

### 4.2. CD24 Affects the Function of Other Molecules

#### 4.2.1. P-Selectin

Leukocyte rolling to sites of tissue destruction and inflammation depends on selectin molecules adhesion mediated by endothelial cells, including I-selectin, E-selectin, and P-selectin [2]. Nevertheless, CD24-surfaced P-selectin also allows tumour cells to interact, causing the rolling of cancerous cells on vascular endothelium, thereby eventually leading to tumour dissemination [41] (Table 1). Further investigations have demonstrated that the sialyl Lex alteration of CD24 is crucial in facilitating the interaction between CD24 and its binding partner, P-selectin [46].

#### 4.2.2. CXCR4

CD24 could act as the “gatekeeper” of the lipid raft [47]. This is demonstrated by CD24’s capacity to modify the localisation and function of other signalling molecules on lipid rafts, including chemokine receptor 4 (CXCR4) [46] (Table 1). Chemokine receptors are identified on the surface of numerous tumour cells. CXCR4, which holds prognostic significance for various cancers [48,49], is the most commonly found of all chemokine receptors in malignant tumours. CXCR4 must be embedded in lipid rafts to achieve maximum efficacy [50]. Nonetheless, a study revealed that CD24 in MDA-MB-231 breast cancer cells limits the CXCR4 localisation in membrane lipid rafts, thereby controlling its downstream signalling changes. It was also observed that CD24-deficient MDA-MB-231 cells improved CXCR4-mediated cell migration [51].

#### 4.2.3. G3BP

As previously discussed, CD24’s function is linked to G3BP, a phosphorylation-dependent ribonucleic acid endonuclease, and BART mRNA in stress granules (Table 1). A study found that BART overexpression inhibited cell invasion and metastasis [14]. Initially, BART was discovered to be a binding partner of ADP-ribosylation factor-like 2 (ARL2), a small G protein that regulates microtubule dynamics and folding. In-depth examination has demonstrated that CD24 impedes the migration of pancreatic ductal adenocarcinoma (PDAC) by binding to G3BP in stress granules and consequently inhibiting the degradation of BART mRNA by the latter [14].

#### 4.2.4. p53

It is widely recognised that p53 was amongst the earliest identified oncogenes; however, it is frequently mutated in human cancers. Alongside p53, ARF has also been found to be an oncogene linked to it, working to protect p53 from degradation via ubiquitination by disabling the E3 ligase MDM2 [17]. The stability of ARF is dependent on its binding to the nucleolar protein NPM. The main cellular functions of NPM are to promote cell proliferation, monitor differentiation, and inhibit programmed cell death. It is widely expressed in rapidly dividing cells. However, CD24 can compete with ARF to bind NPM and promote ARF dissociation from the complex, leading to its degradation [17] (Table 1). As a result, CD24’s role in promoting cancer development via the CD24/NPM-ARF-MEM2-p53 signalling pathway has been demonstrated in prostate cancer [52,53].

### 4.3. The Role of CD24 on Immune Cells

In vivo, immune cells have a crucial function in monitoring and eradicating cancer cells. Nevertheless, tumours can avoid the detection of the immune system via different mechanisms [54]. For instance, numerous cancer cells express CD24 extensively, and CD24 on cancer cells can attach to its ligand, Siglec-10, on multiple immune cells and then emit a “don’t eat me” signal to the innate cancer-suppressing immune cells, particularly macrophages, thereby reducing the immune cells’ phagocytosis of cancer cells. Similar to CD47 and PD-L1, CD24 serves as an innate immune checkpoint in which cancer cells impede the immune function of immune cells and influence the behaviour of specific immune cells, mainly through the CD24-siglec10 pathway. Additionally, many immune cells express CD24 themselves, thus CD24’s existence also influences the proliferation, apoptosis, and function of said immune cells. These immune cells comprise macrophages, T cells, B cells, NK cells, iNKT cells, and others. The subsequent section will elaborate on the effects of CD24 on these immune cells (Figure 4).

#### 4.3.1. Macrophages

One ligand of CD24 is siglec10. CD24, located on the surface of cancer cells, can interact with siglec10 on macrophages’ surface to suppress phagocytosis by macrophages. This pathway is commonly employed by cancer cells for immune evasion [3,55]. An experiment [56] demonstrated that obstructing CD24 with a monoclonal antibody improved macrophages’ ability to phagocytose pancreatic cancer cells (Capan-1 and Panc-1). The phenomenon was also observed in malignancies such as ovarian cancer [3] and breast cancer [3]. In the case of oral squamous cell carcinoma (OSCC), the application of anti-CD24 mAb resulted in the reduction in the number of tumour-associated macrophages (TAMs) [57]. TAMs are macrophages that infiltrate the tumour tissue, and initially thought to act against the tumour, although subsequent research has shown that TAMs primarily promote tumour growth and immunosuppression. CD24 may also modulate the polarisation of TAMs [58]. An investigation has confirmed that alternative activated macrophages (AAMs) are also involved in the regulation of CD24-P-selectin-mediated tumour cell adhesion. This process is connected to the chemokine CCL4, also recognised as the macrophage inflammatory protein (MIP-1β). In the case of high-grade serous ovarian cancer (HGSOC), MIP-β, secreted by AAMs, can activate P-selectin expression and advance the adhesion of CD24 to P-selectin, resulting in HGSOC peritoneal metastasis [43].

#### 4.3.2. T Cells

When lymphocytes are depleted, naive T cells undergo rapid expansion and steady-state proliferation [59]. Ou Li analysed CD24 expression in the steady-state proliferation of T lymphocytes, and confirmed that CD24 expression on T cells is necessary for steady-state T lymphocyte proliferation in a lymphocytopenic environment [60]. As outlined previously, the simultaneous downregulation of F4/80^+^ and CD206^+^ TAMs, in addition to anti-CD24 mAb treatment, led to a noteworthy augmentation in the amount of CD4^+^ and CD8^+^ T cells within the tumour microenvironments (TMEs) of oral squamous cell carcinoma (OSCC). This resulted in the reversal of the immunosuppressive TME [57].

#### 4.3.3. B Cells

The expression of CD24 is high in developing B cells, and was initially believed to be a marker of B cells. However, this expression is not present in mature B cells. As CD24 levels alter during B cell development, it is now utilised as a biomarker for developing B cells [61]. SiglecG/10 is one of the CD24 ligands, an inhibitory receptor on B1 cells, and a siglecG/10 deficiency can result in significant expansion of B cells [62]. Alternatively, CD24 has been demonstrated to encourage apoptosis in B cells that are differentiating early [63,64]. The number of pro-B and pre-B lymphocytes, as a result, was significantly reduced in transgenic mice that overexpressed CD24 [65]. Consequently, CD24 is essential for the development of B cells.

#### 4.3.4. NK Cells

CD24 was minimally expressed on the surface of NK cells, whilst siglec-10 was expressed on NK cells. In comparison with siglec-10^+^ NK cells found in hepatocellular carcinoma (HCC) cells, siglec-10^−^ NK cells secreted lower levels of TNF-α, IFN-g, perforin, and granzyme. This impaired NK effect may be attributed to diminished CD24–siglec10 interactions [66]. This discovery presents a novel mechanism by which tumour cells can evade immunity against NK attack. Anti-CD24 antibody medications obstruct the interaction between the two and could potentially boost the cytotoxic activity of natural killer (NK) cells via this pathway. The accuracy of this hypothesis was confirmed by Yue Han’s research. The laboratory developed the anti-CD24 cG7 antibody. Treatment with cG7 led to increased cytotoxicity and higher secretion of IFN-γ and TNF-α by NK cells, in vitro using Huh-7 and BEL-7402 cells in a dose-dependent manner, and in vivo, resulting in reduced growth of Huh-7 xenografts and improved survival rates in mice, when compared with control groups [67].

## 5. Current Drug Therapy Targeting CD24

Current research into antibody drugs targeting CD24 involves monoclonal antibodies, antibody–drug conjugates (ADCs), and CAR-T and CAR-NK therapies, which we have collated and will outline below. However, as Figure 2 demonstrates, CD24 is extensively expressed in normal tissues and immune cells, and the aforementioned treatment strategies unavoidably attack these cells as they highly express CD24 protein, leading to substantial reduction in the patient’s immune system. Addressing the need for CD24 as a therapeutic target presents a significant obstacle. The current study suggests that maybe a promising approach to optimizing treatment would be to begin with CD24 glycosylation, which varies across different cell types.

### 5.1. Monoclonal Antibodies

As mentioned above, immune cells and cancer cells have direct “don’t eat me” signals, and using antibodies to block this pathway undoubtedly promotes immunity. Here are some anti-CD24 antibodies that cleverly prevent binding between CD24 and siglec-10 by binding to CD24.

#### 5.1.1. ALB9 Antibody

The monoclonal antibody ALB9 targets the CD24 protein core and recognises a short leucine–alanine–proline (LAP) sequence near the CPI anchor site [68]. The effect of the human-specific anti-CD24 mAb ALB9 was investigated by establishing UM-UC-3 human urothelial carcinoma cell tumour models and lung metastasis models. Tumour model data from NCr nu/nu mice showed that ALB9 treatment for 13 days resulted in a lower mean tumour volume than the control saline-treated group, enabled a reduction in the tumour growth rate, and maintained durable inhibition. To determine the effect of ALB9 on established metastatic tumours, tail vein inoculation of lung metastases and treatment with ALB9 after detection of lung metastases in vivo reduced the weight of metastatic tumours and prolonged overall survival by approximately 20% in mice, although metastatic colonisation occurred rapidly in mice after cessation of ALB9 treatment. ALB9 binds to CD24 at a different site from the P-selectin binding site, but still functions in the treatment of metastatic tumours, and therefore ALB9 offers promise for the subsequent treatment of metastatic tumours [15].

#### 5.1.2. SWA11 Antibody

The SWA11 mAb is identical to the ALB9 mAb, binds to CD24 at the core protein, and recognises the same short leucine–alanine–proline (LAP) sequence, and the recognition specificity is independent of the glycosyl chain of CD24 [67,68,69]. The monoclonal antibody SWA11 targeting CD24 was injected intraperitoneally into the A549 human lung cancer model (lacking T cells and B cells, and with NK cells without effector function), the BxPC3 pancreatic cancer model (lacking T cells and B cells), and the SKOV3ip ovarian cancer model (lacking T cells), and a reduction in tumour volume was observed in all cases. In particular, tumour weight in the ovarian cancer model treatment group was reduced by more than 80% compared with the control group [36]. SWA11 mAb treatment by tail vein injection also reduced tumour volume in the HT29 colon adenocarcinoma model in athymic nude mice [70]. U87 glioma xenografts were significantly reduced in volume and weight after treatment with the anti-CD24 monoclonal antibody SWA11 [71]. Unfortunately, these two studies did not test the safety of the SWA11 mAb in the four models mentioned above.

Niko P Bretz investigated the anti-cancer effects of SWA11 mAb in more detail, and found that SWA11 mAb strongly inhibited the growth and proliferation of A549 lung cancer and SKOV3ip ovarian cancer cells, but did not have a major effect on the apoptosis of these two types of cancer cells [36]. Similar conclusions were reached by Eyal Sagiv et al.’s in vitro experiments. They investigated the inhibitory effect of the CD24 monoclonal antibody SWA11 on the growth of the HT29 colon cancer cell line and the Colo-357 pancreatic cancer cell line expressing CD24, and the SW-480 colon cancer cell line, the PANC-1 pancreatic cancer cell line, and the MIA-Paca pancreatic cancer cell line with low or no CD24 expression. Their results showed that the growth of the first two cancer cell lines was strongly inhibited, while the growth of the cell lines with low or no CD24 expression was weakly inhibited or not affected [71,72,73]. In addition, SWA11 mAb treatment of A549 lung cancer and SKOV3ip ovarian cancer also had positive effects on macrophage infiltration and cytokine changes in the tumour microenvironment. Meanwhile, in the A549 lung cancer model, SWA11 mAb pretreatment followed by conventional chemotherapy with gemcitabine was able to slow tumour growth significantly [74]. HT29 colon cancer cells treated with SWA11 mAb also enhanced the therapeutic effect of five chemotherapeutic agents (oxaloplatin, 5-fluorouracil, doxorubicin, irinotecan, and paclitaxel) [70].

#### 5.1.3. Other Monoclonal Antibodies

Data showed that G7 mAb was able to inhibit the proliferation and invasion of A549, HT-29, and Huh-7 tumour cells in vitro. However, the combination of G7 mAb with cetuximab, which binds specifically to cell-surface-expressed EGFR and is one of the most widely studied targeted drugs, was more effective. In vivo, G7 mAb was also able to reduce tumour weight and prolong survival in nude mice, again more effectively in combination with cetuximab. In conclusion, G7 mAb exerted excellent anti-tumour effects in combination with cetuximab. However, in terms of anti-angiogenic effects, cetuximab inhibited the secretion of the angiogenic factor VEGF more than G7 mAb, but, again, the effect was stronger in combination [75].

SN3b is a commonly used antibody to detect CD24. SN3b recognises epitopes that are independent of the CD24 backbone, most likely in the sialic acid-containing portion of the O-strand glycan [68]. SN3b cannot exclude the possibility of cross-reactivity with other possible surface glycoproteins. And due to the highly variable glycosylation of CD24, the glycan epitope is not present on all forms of CD24, so SN3b cannot bind all glycoprotein forms of CD24 [68]. Therefore, SN3b has greater limitations compared with other antibodies that bind to the CD24 core protein.

### 5.2. Antibody–Drug Conjugates

Antibody–drug conjugators (ADCs) have great potential in tumour therapy. Compared with conventional chemotherapy, ADC therapy enhances the selective, targeted delivery of cytotoxic drugs, thereby increasing the therapeutic index of cell therapy drugs. However, as mentioned above, the widespread high expression of CD24 in normal cells/tissues increases the risk of off-target drug delivery.

#### 5.2.1. SWA11-Ricin A Chain Couplers

Ricin A chain immunotoxins (ITs) exert cell toxicity dependent on target antigen-mediated induction into the cytoplasm. The purpose of coupling SWA11 monoclonal antibody to a deglycosylated ricin A chain is to take advantage of the abundant expression of CD24 on cancer cells, so that the deglycosylated ricin A chain, lacking cytoplasmic structural domains, cross-linked to it can be induced into the cell to kill cells. In contrast, the nature of SWA11 binding to the CD24 core protein rather than to peripheral glycosyl groups is more conducive to inducing internalisation of the deglycosylated ricin A chain [76]. Two SWA11-ricin A chain pairs are available, the first being the SWA11-SPDB-dg.ricin A chain constructed by Zangemeister-Wittke for study in small cell lung cancer (SCLC) [76]. Two models were constructed for this study: the first model used traditional S.C. solid-tumour xenografts; the second model simulated the clinical setting of diffuse tumour cells that form the basis of residual disease in SCLC. The SWA11-SPDB-dg.ricin A chain significantly delayed SW2 solid tumour xenograft growth and selectively eliminated clonal SW2 cells in a second model. The second coupler is a coupler that uses the bivalent linker SMPT to link the SWA11 mAb to the deglycosylated ricin A chain—SWA11.dgA.SWA11.dgA exerts significant efficacy in prolonging survival in a diffuse human BL-38 Burkitt lymphoma model [77].

#### 5.2.2. SWA11-Pseudomonas Exotoxin Derivative (PE38)

The construct SWA11-ZZ-PE38, obtained by coupling SWA11 mAb to a Pseudomonas exotoxin derivative (PE38) via the Fc-binding ZZ domain of staphylococcal protein A, significantly reduced the growth of HT-29 colon cancer cell xenografts, while SWA11-ZZ-PE38 was not toxic to normal tissue in mice, but selectively delivered the toxin to CRC cells and caused their apoptosis [78].

#### 5.2.3. hG7-BM3-VcMMAE

The humanised antibody hG7-BM3, combined with the mitogenic toxin MMAE, which plays an effective role in mitotic inhibition by inhibiting tubulin polymerisation, was effective in targeting Huh-7 hepatocellular carcinoma cells in vivo, inhibiting tumour cell proliferation, and inducing apoptosis, resulting in a significant reduction in tumour graft size. The coupling construct was able to enhance the anti-tumour capacity compared with hG7-BM3 or MMAE alone. The study also performed a safety analysis, and the data showed that the hG7-BM3-VcMMAE coupling did not cause weight loss in mice, indicating that the humanised antibody has a good safety profile [79].

#### 5.2.4. G7mAb-DOX

Adriamycin (DOX) is a widely used anti-tumour drug for chemotherapy treatment of many cancers, including liver cancer, but its toxic side effects are excessive. In Zhaoxiong Ma’s study, the high-dose DOX-treated group showed a variety of side effects, such as weakness, poor mobility, depression, and dull, rough fur in mice, and a lower survival rate in the high-dose DOX-treated group than in the low-dose DOX-treated group [80]. However, the data showed that CD24 coupled to DOX ameliorated the toxicity of DOX and, surprisingly, did not alter the binding affinity of the G7 antibody to the antigen and was able to target and inhibit Huh-7 tumour growth in vivo effectively and prolong and increase survival in mice [80].

#### 5.2.5. HN-01

For tumours, an appropriate dose of nitric oxide (NO) is beneficial for their growth. Most studies suggest that high levels of NO have mainly tumour-suppressive effects, while low/medium levels of NO may promote tumour development [81]. Therefore, it has become a challenge to select a suitable carrier to release NO into tumour cells. Fumou Sun developed an ADC called HN-01, which couples the NO donor molecule HL-2 to G7 mAb through a thioether bond. HL-2 could be triggered by high levels of glutathione (GSH) in tumour cells to release NO in situ. HN-01 was able to inhibit tumour growth effectively in mice inoculated with the liver cancer cells BEL-7402, Huh-7. At the same dose, HN-01 treatment had greater anti-tumour efficacy than G7 mAb co-treatment with HL-2 (non-coupled situation) [79].

### 5.3. CAR-T

Chimeric antigen receptor (CAR) T-cell therapy involves combining T cells with a synthetic CAR receptor that has been engineered to target cells that express a specific antigen. There is limited research on CAR-T therapies targeting CD24. Anti-CD24 CAR-T appears to be the first to make good progress in the treatment of pancreatic cancer. Anti-CD24 CAR-T was injected into pancreatic cancer tumours by tail vein injection or directly into pancreatic cancer tumours accompanied by intraperitoneal IL-2 injections twice a day for ten days by Amit Maliar et al. The results showed good performance of anti-CD24 CAR-T against tumours. Although only a small percentage of tumour cells expressed CD24, the CD24-targeted CAR-T was still effective in prolonging survival in mice [82].

### 5.4. CAR-NK

Similar to the manufacturing principle of CAR-T, CAR-NK also refers to the cells that result from the clever combination of engineered CAR and NK cells. Rüdiger Klapdor designed a novel third-generation CAR-NK targeting CD24 for the treatment of ovarian cancer, and they selected a single-chain variable fragment (scFv) in SWA11 mAb to be placed on the third-generation CAR backbone. The novel anti-CD24-CAR-NK-92 cells showed the ability to eliminate ovarian cancer SKOV3 and OVCAR3 cells, which express high levels of CD24, completely when co-incubated with them. Surprisingly, the engineered NK-92 cells also exhibited self-renewal behaviour after killing target cells, as observed by fluorescence microscopy [83].

### 5.5. Clinical Research

As mentioned above, CD24-targeted therapies, including monoclonal antibodies, ADCs, and cellular immunotherapy, have shown promising anti-tumour activity in preclinical studies. However, clinical trials targeting CD24 are still limited (Table 2). The first two provided clinical data on the use of the recombinant fusion protein CD24Fc for the treatment of advanced solid tumours and melanoma (NCT04552704 and NCT04060407); the NCT04060407 trial was terminated early and the NCT04060407 trial was withdrawn before enrolment. The last two trials (NCT05985083 and NCT06028373) are anti-CD24mab trials. In one of these trials (NCT05985083), IMM47 is a humanised mAb against CD24 that is administered by intravenous infusion every 2 weeks in 28-day treatment cycles. The other trial (NCT06028373) is still recruiting.

## 6. Risks and Opportunities of Anti-CD24 Drugs in Cancer Treatment Process

### 6.1. CD24 Plays a Role in Tumour Drug Resistance

During clinical treatment, it remains important to consider the impact of CD24 on drug resistance. Several studies have shown that CD24 expression appears to be positively correlated with chemoresistance [84]. RNA sequencing and proteomic analysis of cisplatin-resistant and cisplatin-sensitive head and neck squamous cell carcinoma (HNSCC) identified CD24 as one of the potential genes associated with drug resistance in HNSCC. Meanwhile, several preclinical data showed that CD24 overexpression in HNSCC was associated with chemotherapeutic drug resistance [85]. Experiments have also shown significant differences in CD24 expression between drug-resistant and drug-sensitive cells in NSCLC. CD24 overexpression decreases the sensitivity of melanoma to BRAF inhibitors via the src/STAT3 pathway [86]. Overexpression of CD24 in hepatocellular carcinoma cells increases sorafenib resistance by activating autophagy in HCC [87]. CD24^+^ CAOV3 and TOV21G ovarian cancer cell lines were found to be more resistant to cisplatin and adriamycin [88]. Chao Chen’s study also showed that CD24 can accelerate immune evasion and increase resistance to chemotherapeutic drugs by upregulating siglec-10 in ovarian cancer [89]. CD24^+^ cells were found to be enriched in gemcitabine-resistant pancreatic cancer cells [90]. Artemin (ARTN) reduces sensitivity to doxorubicin and paclitaxel by upregulating CD24 expression in endometrial cancer cells [91]. Therefore, therapeutic strategies that target CD24, including reducing the ability of cancer cells to express CD24 or targeting cancer cells with high CD24 expression, are expected to improve the clinical problem of drug resistance.

### 6.2. CD24 Prevents Excessive Inflammatory Response

Under physiological conditions, SiglecE binding to sialylated CD24 has been reported to induce recruitment of the protein tyrosine phosphatase SHP-1. SHP-1 can inhibit cell proliferation and inhibit excessive inflammation to prevent the development of metabolic syndrome, and inactivation or blockade of the CD24-SiglecE pathway will exacerbate diet-induced metabolic disorders, including obesity, dyslipidaemia, insulin resistance, and non-alcoholic steatohepatitis (NASH) [92]. The study also raised the hypothesis that the CD24-Siglec-E axis in macrophages is responsible for their suppression of excessive inflammation and metabolic disorders, but more evidence is needed to prove this hypothesis [92].

Pathogen-associated (PAMP) and/or danger-associated (DAMP) molecular patterns are two major inducers of inflammation, but DAMP, such as high-mobility group protein 1, heat shock proteins, and nucleophosmin, can cause limited inflammation. The CD24-siglecG (mouse)/10 (human) pathway inhibits the host response to DAMP [93,94]. Therefore, disruption of the CD24-siglecG/10 pathway may cause an excessive inflammatory response. Follow-up experiments by Guoyun Chen [95] confirmed this claim by targeting mutations in CD24 and siglec G, resulting in an increased risk of sepsis due to bacterial infection. Therefore, it is important to pay attention to the prevention and suppression of the body’s excessive inflammatory response caused by antibody drugs during oncological treatment.

## 7. Conclusions

In this review, we provide an overview of the progress made in the use of antibodies targeting CD24 as an anti-cancer therapy. CD24, a GPI-anchored glycoprotein, has been shown to interact with several signalling and cross-linking molecules, contributing to cancer progression. In addition, CD24, similar to CD47 and PD-L1 checkpoints, can reverse the function of immune cells, resulting in overall pro-tumour progression.

Ongoing clinical trials have shown promising results in the development of antibody drugs that block CD24 ligand binding. To date, monoclonal antibodies, antibody–drug conjugates, and cellular therapies such as CAR-T and CAR-NK have shown significant efficacy. However, clinical data on anti-CD24 antibody drugs remain limited.

Compared with alternative targets, CD24 appears to be a favourable choice, mainly because of its expression pattern. During the investigation, CD24 was found to be highly expressed universally in various tumours, its glycosylation patterns were very different in cancer cells and normal cells, and CD24 was increasingly correlated with CSCs. With this series of important discoveries, the mystery of CD24 seems to be gradually unravelling. However, CD24 also poses challenges and risks in target development and design. CD24 is highly expressed not only in various cancer cells but also in individual normal tissues such as the oesophagus and thyroid. Most importantly, the glycosylation pattern of CD24 remains elusive, posing a significant challenge for antibody development. Meanwhile, tumour resistance and the risk of inflammation are also issues that need to be addressed in CD24 antibody drug therapy.

In conclusion, there remains significant untapped potential in the development of CD24 antibody drugs, and we believe there is an urgent need to find the unique glycosylation pattern in cancer cells in order to design antibody drugs targeting this specific site of CD24, which will require concerted efforts to develop effective antibody drugs.

## Figures and Tables

**Figure 1 biomedicines-11-03159-f001:**
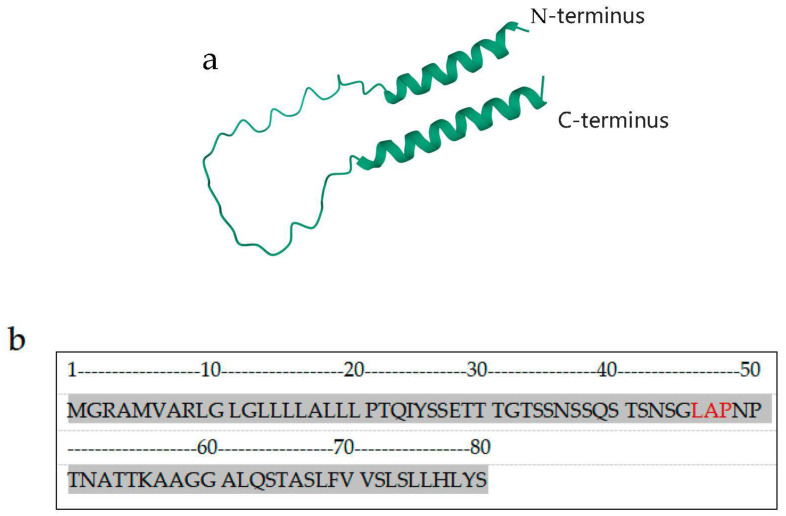
Structure of human CD24. (**a**) Schematic showing the predicted peptide structure of human CD24 from NCBI (Protein code: P25063), showing the N-terminus of the peptide chain at the top and the C-terminus of the peptide chain at the bottom. (**b**) Schematic showing the primary structure of the CD24 protein: the first amino acid is the N terminal, the 80th amino acid is the C terminal, and the red letters are the LAP sequence of the binding site of the anti-CD24 antibodies ALB9 and SWA11.

**Figure 2 biomedicines-11-03159-f002:**
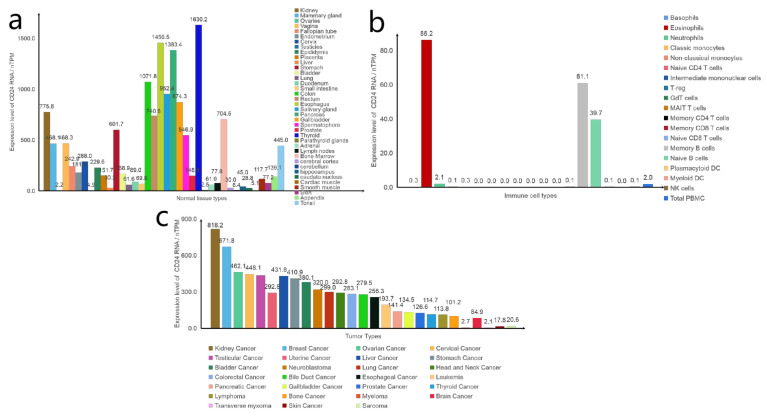
Expression levels of CD24 RNA. (**a**) Expression levels of CD24 RNA in different normal tissues. (**b**) Expression levels of CD24 RNA in different immune cells. (**c**) Expression levels of CD24 RNA in different tumours. Data from the “The Human Protein Atlas” database.

**Figure 3 biomedicines-11-03159-f003:**
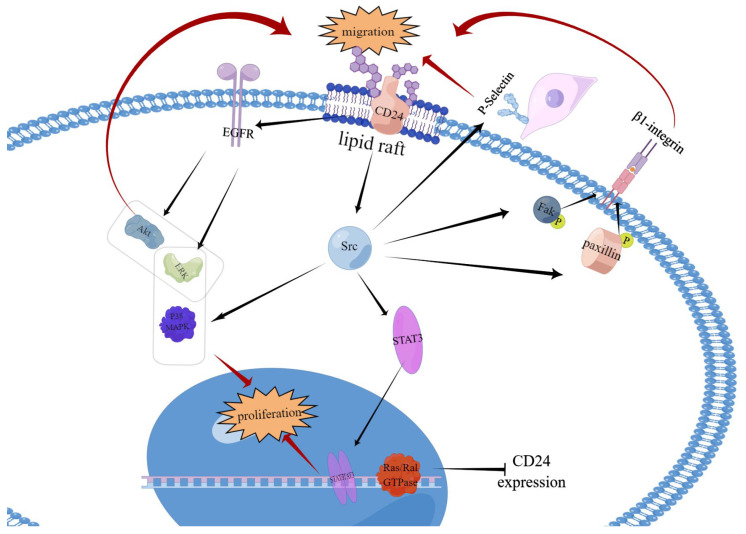
Interaction of CD24 with signalling molecules. The interaction between CD24 and P-selectin facilitates the rolling of specific cancer cells on endothelial cells, as well as their adhesion to platelets or mesothelial cells. It thereby promotes the migration and spread of cancer cells. Within intact lipid rafts, CD24 interacts with c-Src, promoting its activity without altering its expression level. Enhanced Src activity accelerates the phosphorylation of its substrates FAK and paxillin. This enhances integrin-mediated adhesion, indirectly stimulating cell adhesion to fibronectin, collagen I and IV, and laminin, through the activation of α3β1 and α4β1 integrin. This promotes cancer cell metastasis and invasion. Src activated by CD24 tyrosine phosphorylates STAT3 signalling molecules, leading to their activation. Upon activation, STAT3 dimerizes and migrates to the nucleus, where it specifically binds to DNA to facilitate target gene expression. CD24 prevents EGFR internalisation and degradation by means of RhoA. Two critical downstream pathways of EGFR are the PI3K/Akt and ERK pathways, which are associated with cancer cell growth and migration. The activation of Akt and/or ERK can cause E-calmodulin to be downregulated, promoting cancer cell migration. CD24 is a downstream protein of Ras, as demonstrated by previous studies. It has been revealed that activated Ras can repress CD24 expression, although the picture has not been shown. The proliferation induced by CD24 is reliant on the activation of ERK1/2 and p38 MAPK within colorectal cancer. However, it has no impact on proliferation through JUK.

**Figure 4 biomedicines-11-03159-f004:**
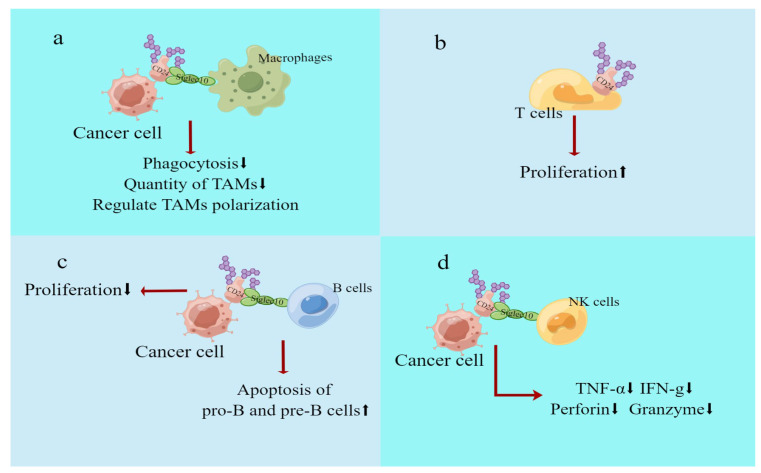
Effect of CD24 on immune cells. (**a**) CD24 on the surface of cancer cells can interact with siglec10 on the surface of macrophages to inhibit phagocytosis of macrophages. CD24 may also regulate TAM polarisation and quantity. (**b**) CD24 expression on T cells is necessary for the steady-state proliferation of T lymphocytes in a lymphocytopenic environment. (**c**) CD24 is now widely used as a biomarker for developing B cells. (**d**) Compared with siglec-10^+^ NK cells in hepatocellular carcinoma (HCC) cells, siglec-10^−^ NK cells secreted less TNF-α, IFN-g, perforin, and granzyme, and this impaired NK effect may be due to reduced CD24–siglec10 interactions. The red arrow indicates the result of the action, and the black arrow indicates up or down.

**Table 1 biomedicines-11-03159-t001:** CD24 and its partners.

Name	Function	Location
P-Selectin	Promotes cell adhesion	Cytoplasmic membrane
CXCR4	Promote cell migration	Raft region
G3BP	Inhibit cell migration	Cytoplasm/cell membrane
P53	Oncogenes, mutations that cause cells to become cancerous	Nucleus (possible)

**Table 2 biomedicines-11-03159-t002:** CD24 and clinical research.

	ID	Drug Type	Disease Type	Clinical Progress
1	NCT04552704	CD24 Fc	Advanced Solid Tumours	Phase I/II
2	NCT04060407	CD24 Fc	Metastatic Melanoma	Phase Ib/II
3	NCT05985083	IMM47	Advanced Solid Tumours	Phase I
4	NCT06028373	ATG-031	Advanced Solid Tumours or B-cell Non-Hodgkin Lymphomas	Phase I

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
