# Peer review of "Targeting CD24 in Cancer Immunotherapy"

_biomedicines, 2023, doi:10.3390/biomedicines11123159_

Round 1
Reviewer 1 Report
Comments and Suggestions for Authors
Thank you for the opportunity to review this article by Chen et al on CD24 as a potential therapeutic target in cancer. The article provides a useful reference for studies in the field. I have a number of comments:
· A more critical view of published data and how they should be interpreted is warranted. Just to provide a few examples:
· Section 3.2. is entitled ‘CD24 is a marker of CSC’; however, the data discussed suggest this is not accurate.
· Section 4.1.2. – correlation does not equal causation
· Section 5.1: for the in vivo studies described, what is the proposed therapeutic mechanism of action? Are immune competent animals used in these studies? Are the antibodies acting via direct effects on cancer cells or via immune modulation?
· Section 4.3.2 What is the proposed mechanism of action of the CD24 antibody described?
· Section 6.1. – through what mechanism of action could CD24 antibodies be leveraged to alleviate drug resistance?
· There are considerable challenges to developing CD24 as a therapeutic target and it would be helpful for the published studies to be placed in this context. For example, in section 5 where ADC and CAR-T cell approaches are described it would be good to discuss in the context of potential toxicity. CD24 has wide expression in normal tissues and immune cells; what would be the consequences of killing these cells? Do the ADC and cell therapies tried in these pre-clinical models cross-react with the mouse target to enable assessment of risk? How could CD24 on cancer cells be targeted more selectively? This latter point is referred to at the end of the conclusion but would benefit the discussion if considered earlier.
· A section on CD24 focused clinical studies should be included. What approaches have been taken in the clinic, and what data is available that could help future development and indicate feasibility of targeting this pathway. (Of note in the context of toxicity, trials are underway to test the effectiveness of a CD24-Fc to alleviate toxicities associate with other immune therapies).
· Figure 1 needs some annotation. For example which are the C- and N- termini and where are the attachment sites for O- and N-linked glycans? A suggestion would also be to include the peptide sequence as it is very short; in this case the LAP sequence that is the binding site of two CD24 experimental antibodies described could be indicated
· Figure 3: needs a legend explaining the interactions and pathways shown.
· 3.1.1. Subsection needs a title
· Line 203: inhibit phagocytosis BY macrophages
· Line 239 – what is cG7?
Comments on the Quality of English LanguageGood quality English is used
Author Response
Thank you for your valuable advice. I have revised it. Please see the attachment for details

Reviewer 2 Report
Comments and Suggestions for Authors
In this review the authors exhaustively describe the CD24 protein, its structure, its biological role, its expression in various types of tumors, its interaction with other signaling pathways and finally all the classes of drugs that are used as its inhibitors. In my opinion, CD24 as a tumor target is well described in this review and provides scientifically useful information to the scientific community interested in this topic. The review is well written and organized in all its sections, always keeping its focus and the references are correct.
Author Response
Thank you for your kind words. Thank you for your comments.
Reviewer 3 Report
Comments and Suggestions for Authors
1. Figure 2 captions need to be written properly in the paragraph style.
2. Add more details in the caption of Figure 3. All figures should be independently understood.
3. Add more details in the caption. of Fig. 4
Comments on the Quality of English LanguageIt's fine
Author Response

(The authors gave the same response as above.)

Reviewer 4 Report
Comments and Suggestions for Authors
Dear Zhigang Guo and colleagues,
I read your review on CD24 entitled “Targeting CD24 in cancer immunotherapy” with interest.
Here are my comments:
The article is written for an audience already familiar with the topic and its technical terms. As this narrows the outreach of the manuscript, it will be important to edit the text such that it becomes more accessible for a wider readership. In general, all abbreviations need to be fully explained. I will point this out below.
P1, L20 glycoprotein of the plasma membrane where is it is found in lipid rafts.
I think here would be a good point to tell the reader how CD24 can come into contact with intracellular targets like G3BP in stress granules or nucleolar protein NPM maybe within the cytoplasm. This is not yet clear in the text. It may help to include an image or table showing the different locations of CD24 and its interaction partners. It needs to be explained to the reader how a membrane protein like CD24 can exert interactions with intracellular proteins.
P1, L39 what does HSA mean after CD24?
P2, L55 delete 3.1.1 Subsubsection
Figure 1 Please include Protein Accession Code: P25063
Figure 2b the colour choice is not good as the three bars are all of similar shades of blue-green and therefore difficult to be distinguished
Figure 3 please show CD24 with its lipid anchor – currently it looks like an integral membrane protein
P3 L73 please use full name of CSCs (please do not use abbreviations in headings & subheadings)
P5 L132 full name of EGFR please
As a general comment, please introduce briefly the roles of the proteins discussed like BART, NPM, MIP-beta, F4/80+, TAMs). The reader may not know about them.
P6 L 182 please explain PDAC
P6 L 193 The Role of CD24 as immune cell inhibitor
P6 L198 please explain here in more detail the relationship between CD24 and siglec10 as this is very important for the rest of the story. Please explain the roles of inhibitory receptor sialic-acid-binding Ig-like lectin 10 (Siglec-10) here in more detail. Especially on which cell types this protein is expressed (ie macrophages).
P6 l 227 a Table listing the discussed CD24 ligands would be good (plus their role and localisation). Please say that CD24 is not expressed on mature B cells.
P7 L239-241 the sentence starting “cG7 treatment…” is difficult to understand. Please introduce first the cG7 antibody
Figure 4 please use panel letters (ie 4A, 4B, 4C, 4D)
P8 L266 better SWA11 Antibody (same for the other related subheadings)
P8 L269 what are the other glycans?
P8 L300 please introduce the antibody cetuximab
P9 5.2 please use full name of ADC
P9 L338 please introduce NMAE
P10 L 360 please introduce HL-2
P10 L366 please introduce CAR-T cell therapy
P10 L374 please introduce CAR-NK
P10 L377 please introduce SKOV3 & OVAC3 cells
P10 L404 please introduce SHP-1 recruitment
Comments on the Quality of English LanguageThe English needs a moderate editing to improve the fow and to correct some of the wording
Author Response

(The authors gave the same response as above.)

Round 2
Reviewer 1 Report
Comments and Suggestions for Authors
Thank you for the opportunity to re-review this manuscript. The authors have addressed my concerns.
Comments on the Quality of English LanguageSome minor editing of English language will be required.
Author Response
Thank you for your reply and your suggestions!
Reviewer 4 Report
Comments and Suggestions for Authors
Dear Wenwen Chen, Zhigang Hu and Zhigang Guo,
thank you very much for your thorough editing of the manuscript. It reads now much better.
Here are my last minor comments:
Figure 1: Please use the Protein code P25063 and please say that this is human CD24. Regarding panel B, it may look better to show only the amino acid single letter code for CD24 with the AB binding site highlighted rather than the screen shot. You can make yor own figure.
L 79 the localisation in the cytoplasm is probably in cytoplasmatic vesicles – (please see https://www.proteinatlas.org/ENSG00000272398-CD24/subcellular#human).
Comments on the Quality of English LanguageSome final editing is needed
Author Response
Thank you for carefully giving us valuable suggestions again. Thanks for your comments, we have made corresponding modifications, please see the attachment for details.
